# Microvascular Complications and Cancer Risk in Type 2 Diabetes: A Population-Based Study

**DOI:** 10.3390/cancers17111760

**Published:** 2025-05-23

**Authors:** Yu-Hsin Yen, James Cheng-Chung Wei, Fu-Shun Yen, Yung-Shuo Kao, Heng-Jun Lin, Der-Yang Cho, Chii-Min Hwu, Chih-Cheng Hsu

**Affiliations:** 1Duke-NUS Medical School, 8 College Rd., Singapore 169857, Singapore; yvonneyen26@gmail.com; 2Department of Allergy, Immunology & Rheumatology, Chung Shan Medical University Hospital, No. 110, Sec. 1, Jianguo N. Rd., South District, Taichung 40201, Taiwan; jccwei@gmail.com; 3Graduate Institute of Integrated Medicine, China Medical University, No. 91, Hsueh-Shih Road, Taichung 40402, Taiwan; 4Institute of Medicine, Chung Shan Medical University, No. 110, Sec. 1, Jianguo N. Rd., South District, Taichung 40201, Taiwan; 5Dr Yen’s Clinic, No. 15, Shanying Road, Gueishan District, Taoyuan 33354, Taiwan; yenfushun@gmail.com; 6Department of Radiation Oncology, Taoyuan General Hospital, Ministry of Health and Welfare, Taoyuan 330215, Taiwan; codingforlifetime@gmail.com; 7Management Office for Health Data, China Medical University Hospital, China Medical University, Taichung 404327, Taiwan; hengjun.cmuh@gmail.com; 8Translational Cell Therapy Center, Department of Medical Research, China Medical University Hospital, Taichung 40447, Taiwan; 9Department of Neurosurgery, China Medical University Hospital, Taichung 40447, Taiwan; 10Graduate Institute of Biomedical Sciences, China Medical University, Taichung 40447, Taiwan; 11Faculty of Medicine, National Yang-Ming Chiao Tung University School of Medicine, No. 155, Sec. 2, Linong Street, Taipei 11221, Taiwan; 12Section of Endocrinology and Metabolism, Department of Medicine, Taipei Veterans General Hospital, No. 201, Sec. 2, Shipai Road, Beitou District, Taipei 11217, Taiwan; 13Institute of Population Health Sciences, National Health Research Institutes, 35 Keyan Road, Zhunan 35053, Taiwan; 14Department of Health Services Administration, China Medical University, No. 91, Hsueh-Shih Road, Taichung 40402, Taiwan; 15Department of Family Medicine, Min-Sheng General Hospital, 168 ChingKuo Road, Taoyuan 33044, Taiwan; 16National Center for Geriatrics and Welfare Research, National Health Research Institutes, No. 8, Xuefu W. Rd., Huwei 632007, Taiwan

**Keywords:** type 2 diabetes, microvascular complications, diabetic kidney disease, diabetic retinopathy, diabetic neuropathy, cancer incidence, cancer mortality

## Abstract

Microvascular complications in type 2 diabetes and cancer share common biological pathways. Our study revealed that while microvascular complications are associated with increased all-cause mortality in type 2 diabetes, they do not elevate the risk of cancer or cancer-specific mortality. These findings offer important epidemiological insights into the connection between diabetes-related complications and cancer.

## 1. Introduction

Type 2 diabetes mellitus (T2D) is a prevalent metabolic disorder characterized by insulin resistance and chronic hyperglycaemia, leading to various complications [1,2,3]. Among these, microvascular complications—such as diabetic nephropathy, retinopathy, and neuropathy—are major contributors to morbidity and mortality [4,5,6]. Notably, diabetic nephropathy and retinopathy affect approximately 25% of patients with T2D, while diabetic neuropathy occurs in nearly 50% of this population [7].

In addition to vascular complications, T2D is associated with an increased risk of developing various cancers, including liver, pancreas, colorectal, endometrial, and breast cancers [8,9,10]. The proposed mechanisms linking T2D to cancer include chronic hyperinsulinaemia, low-grade systemic inflammation, increased oxidative stress, and abnormalities in insulin-like growth factor (IGF) signalling and angiogenesis [11,12,13]. Emerging evidence suggests that microvascular complications may independently contribute to cancer development. Chronic hyperglycaemia—the primary driver of microvascular disease—induces endothelial dysfunction, advanced glycation end-product (AGE) accumulation, persistent oxidative stress, and impaired tissue perfusion [14]. These same pathophysiological processes are implicated in tumorigenesis by promoting DNA damage, cellular proliferation, and abnormal angiogenesis [15,16]. For example, diabetic nephropathy has been linked to increased circulating pro-inflammatory cytokines and impaired immune surveillance, potentially facilitating cancer development [17,18]. Retinal ischaemia and microvascular remodelling in diabetic retinopathy have also been associated with systemic vascular dysfunction, which may reflect a broader milieu conducive to oncogenesis [19]. Despite these plausible mechanistic connections, the independent role of microvascular complications in cancer risk and mortality remains poorly understood. Few studies have examined this relationship in depth, and most have been limited to specific cancer types or diabetic complications [17]. Furthermore, the current literature lacks large-scale, population-based analyses that stratify cancer risk by the burden of microvascular disease. These data gaps exist not only in Taiwan but also globally, highlighting a broader need for research in diverse populations.

To address this, we utilized the Taiwan National Health Insurance Research Database (NHIRD), a comprehensive, nationwide dataset, to evaluate the association between microvascular complications and cancer outcomes in patients with newly diagnosed T2D. Specifically, we assessed whether the presence—and cumulative number—of microvascular complications (diabetic kidney disease, retinopathy, and neuropathy) was associated with increased cancer incidence and cancer-related mortality. By clarifying these associations, our study aims to enhance the understanding of the vascular-cancer axis in T2D, inform cancer risk stratification, and support the development of targeted surveillance strategies to improve long-term outcomes.

## 2. Materials and Methods

### 2.1. Study Population and Data Source

Taiwan’s National Health Insurance (NHI) programme was launched in 1995 and achieved near-universal coverage by 2000, now enrolling about 99% of the country’s 23 million residents. Patient records from this system are stored in the NHIRD, which contains comprehensive healthcare data, including demographics, diagnoses, treatments, and hospitalisations, all coded using ICD-9/10-CM standards. The NHIRD is also linked to the National Death Registry for accurate cause-of-death verification. For this study, we used anonymized, encrypted data from the NHIRD to ensure patient and provider confidentiality [20]. Ethical approval was obtained from the Research Ethics Committee of China Medical University and Hospital [CMUH110-REC1-038 (CR-3)], with a waiver for informed consent due to the de-identified nature of the data.

### 2.2. Study Procedures

This study followed individuals newly diagnosed with T2D between 1 January 2008 and 31 December 2021, tracking their outcomes until 31 December 2021. T2D was identified using ICD codes (Appendix A), requiring either three or more outpatient visits within a year or a single hospitalisation. This approach has been validated in Taiwan, demonstrating 93.3% accuracy [21]. To ensure a well-defined study cohort, several exclusion criteria were applied: patients were excluded if they had missing data within the first year of T2D diagnosis, a pre-existing history of diabetic kidney disease, diabetic retinopathy or diabetic neuropathy before T2D diagnosis, an index date outside the 2009–2018 range, a diagnosis of type 1 diabetes, or were younger than 18 or older than 80 years. Additionally, patients were excluded if outcomes occurred before the index date or if their follow-up period was less than 180 days.

In the first year after T2D diagnosis, patients were categorised based on the presence of microvascular complications. Those without any complications were placed in a non-microvascular disease group, while those with microvascular complications were further stratified. The microvascular disease group was classified into three subgroups: (1) patients with one microvascular disease, which included diabetic kidney disease (DKD) (e.g., diabetic and hypertensive nephropathy, proteinuria, chronic kidney disease, dialysis, renal failure, and renal replacement therapy), diabetic retinopathy (DR) (e.g., background and proliferative retinopathy, retinal oedema and detachment, macular degeneration, vitreous haemorrhage, and vision loss), and diabetic neuropathy (DN) (e.g., mononeuropathy, polyneuropathy, neuralgia, cranial and peripheral nerve palsies, autonomic neuropathy, and Charcot’s disease); (2) patients with two microvascular diseases; (3) patients with three microvascular diseases. To standardise follow-up timing and outcome assessment, the index date was set as the 366th day after T2D diagnosis, ensuring all patients had a full year for microvascular complication assessment before cancer and mortality outcomes were analysed.

### 2.3. Baseline Characteristics and Medications

This study accounted for multiple potential confounders that could impact outcomes, including gender, age, obesity, smoking status, and key comorbidities such as hypertension, dyslipidaemia, coronary artery disease, stroke, heart failure, atrial fibrillation, peripheral arterial disease, chronic obstructive pulmonary disease (COPD), alcohol-related disorders, liver cirrhosis, and connective tissue disorders. Psychiatric and oncological factors were also considered, including family history of neoplasm, benign neoplasms, psychosis, major depressive disorder, and dementia [22]. Additionally, the analysis incorporated data on oral antidiabetic medications, insulin, glucagon-like peptide-1 receptor agonists (GLP-1 RAs), the number of antihypertensive drugs prescribed, and the use of aspirin and statins.

### 2.4. Main Outcomes of Interest

This study examined the incidence of newly diagnosed neoplasms, including all cancer, cancers of the oral cavity, breast, respiratory system, digestive system, lymphoid tissues, and male and female genital organs, as well as cancer-related mortality and all-cause mortality. These outcomes were identified using ICD codes, requiring either at least two outpatient diagnoses within one year or a single hospitalisation, with verification through medication records (Appendix A). Participants were followed until the first occurrence of any of these outcomes, death, or the study’s end date (31 December 2021), whichever came first.

### 2.5. Statistical Analyses

Statistical analyses were conducted to compare baseline characteristics and evaluate differences in outcome between groups. Student’s *t*-test was used to assess differences in continuous variables, such as mean age, while chi-squared tests were applied for categorical variables to compare patients with and without microvascular complications. To minimize confounding, 1:1 propensity score matching (PSM) was performed, pairing patients with and without microvascular disease, as well as those with different combinations of diabetic kidney disease, diabetic retinopathy, and diabetic neuropathy (Appendix A). Matching was based on baseline demographics, comorbidities, and medication use, with a standardised mean difference (SMD) below 0.1 considered indicative of well-balanced groups.

To assess outcome risks, Cox proportional hazards models were used, adjusting for key variables such as age, gender, obesity, smoking status, pre-existing conditions, and medication use. The proportional hazards assumption was verified using proportional hazards tests and Schoenfeld residuals. Hazard ratios (HRs) with 95% confidence intervals (CIs) were reported to quantify risk differences. The Kaplan–Meier plots were generated to visualise cumulative incidence trends for major organ cancers, cancer-related deaths, and all-cause mortality across the study groups. A two-tailed *p*-value < 0.05 was considered statistically significant. All analyses were conducted using SAS version 9.4 and RStudio version 4.4.1.

## 3. Results

Among patients newly diagnosed with T2D, we assessed the prevalence of microvascular complications occurring within the first year after diagnosis. Between 2008 and 2021, a total of 3,501,112 individuals with T2D were identified. After applying exclusion criteria, the following patients were removed: 213,893 who lacked follow-up data within the first year post diagnosis; 609,064 with a pre-existing history of diabetic kidney disease, diabetic retinopathy, or diabetic neuropathy; 606,290 diagnosed outside the 2009–2018 period; 6922 with type 1 diabetes; 125,843 outside the 18–80 age range; and 111,033 who had cancer outcomes prior to the index date. Additionally, 18,570 patients were excluded due to death or a follow-up shorter than 180 days. After these exclusions, the final study cohort consisted of 1,809,497 patients, including 1,414,677 without microvascular complications and 394,820 with at least one microvascular complication (Figure 1).

To reduce confounding, 1:1 propensity score matching was performed based on sex, age, comorbidities, and medication use, resulting in 387,632 matched patients in both the microvascular and non-microvascular groups (Figure 1, Appendix A). Among patients with microvascular complications, additional 1:1 propensity matching with corresponding non-microvascular controls yielded 340,269 individuals with one microvascular disease, 43,109 with two microvascular diseases, and 4254 with three microvascular diseases (Figure 1, Appendix A). The average follow-up duration for the cohort was 8.56 years.

Our analysis found that microvascular complications in T2D did not significantly impact overall cancer risk, with an adjusted hazard ratio (aHR) of 1.00 (95% CI: 0.98–1.01). Similarly, the risks of specific cancers—including oral cavity cancer [aHR 0.99 (95% CI: 0.94–1.05)], thyroid cancer [aHR 1.07 (95% CI: 0.96–1.19)], breast cancer [aHR 0.95 (95% CI: 0.91–1.00)], respiratory organ cancer [aHR 0.97 (95% CI: 0.93–1.01)], digestive organ cancer [aHR 0.99 (95% CI: 0.97–1.01)], lymphoid tissue cancer [aHR 1.02 (95% CI: 0.95–1.09)], and genital organ cancer [aHR 0.99 (95% CI: 0.96–1.03)]—did not differ significantly between patients with and without microvascular disease (Table 1). Further stratification showed that cancer risk remained unchanged across categories of patients with one, two, or three microvascular diseases, with no evidence of a dose-dependent effect (Table 2). However, patients with two microvascular complications were associated with a significantly higher risk of lymphoid tissue cancers [aHR 1.24 (95% CI: 1.02–1.52)] (Table 2). Likewise, cancer-related mortality did not differ between those with and without microvascular disease [aHR 1.00 (95% CI: 0.98–1.02)] (Table 1).

The cumulative incidence of all cancers, as well as site-specific cancers involving the oral cavity, thyroid, breast, respiratory organs, digestive organs, lymphoid tissues, and genital organs, and cancer-related death, did not differ significantly between patients with and without microvascular disease over the 13-year follow-up period (Figure 2). Similarly, there were no significant differences in the cumulative incidence of all cancers or cancer-related death among patients with one, two, or three microvascular complications compared to those without microvascular disease (Figure 3).

While microvascular disease was not associated with an increased risk of cancer or cancer-related death, it was significantly linked to higher all-cause mortality in a dose-dependent manner. The risk of all-cause mortality increased with one microvascular disease [aHR 1.16 (95% CI: 1.15–1.17)], two microvascular diseases [aHR 1.42 (95% CI: 1.38–1.45)], and three microvascular diseases [aHR 1.71 (95% CI: 1.60–1.83)] (Table 2). This dose-dependent effect was further reflected in the cumulative incidence of all-cause mortality, with log-rank *p*-values < 0.001 for one, two, and three microvascular diseases compared to no microvascular disease (Figure 3).

## 4. Discussion

In this nationwide cohort study, we examined the relationship between microvascular complications in type 2 diabetes and subsequent cancer risk, as well as their impact on cancer-related and all-cause mortality. Our findings suggest that microvascular disease does not significantly increase the risk of developing cancer or cancer-related mortality. This held true across multiple cancer types, including those of the oral cavity, thyroid, breast, respiratory, digestive, lymphoid, and genital organ. Furthermore, we observed no dose-dependent relationship between the burden of microvascular complications and the incidence of cancer or cancer-related death. An exception was noted in patients with two microvascular complications who had a significantly higher risk of lymphoid tissue cancers (Table 2); however, this association was not observed in patients with one or three microvascular complications. Similarly, no significant association was found when comparing the presence versus absence of any microvascular complication in relation to lymphoid tissue cancer risk. The basis for this isolated finding remains unclear and warrants further investigation in future studies. However, we observed a clear dose-dependent increase in all-cause mortality with the presence of one, two, or three microvascular complications. This association was more pronounced and specific than previously reported [23], highlighting the substantial impact of microvascular disease on overall survival in T2D patients.

Although microvascular disease and cancer share common underlying mechanisms, including chronic inflammation, oxidative stress, and angiogenic dysregulation, these parallel processes may not necessarily translate into a causal relationship. One possible explanation for the lack of association is that microvascular dysfunction, while contributing to endothelial damage and impaired tissue perfusion, may not directly facilitate tumourigenesis. Instead, both conditions may arise independently as complications of long-standing metabolic dysfunction in T2D. Furthermore, although microvascular complications in T2D involve excessive angiogenesis in certain tissues, such as the retina, they are paradoxically associated with impaired angiogenesis in peripheral tissues—a process critical for both tissue repair and tumour growth [24,25]. While excessive angiogenesis supports tumour progression, inadequate angiogenic response in microvascular disease could theoretically restrict tumour vascularisation, limiting tumour development [26,27,28]. This complex interplay may explain why microvascular disease does not appear to significantly increase cancer risk.

Another important consideration in interpreting our findings is the potential influence of competing risks, particularly mortality related to microvascular complications. In our study, the mean age of the cohort was 58.91 years in the non-microvascular group and 58.74 years in the microvascular group, with long-term follow-up over 13 years. T2D patients with severe microvascular disease often have higher rates of cardiovascular disease, infections, and kidney failure, leading to increased early mortality [29]. This may reduce the likelihood of cancer diagnosis over time, effectively masking any potential long-term relationship between microvascular complications and malignancy. Patients who succumb to other diabetes-related complications may not survive long enough for cancer to develop or be detected, potentially underestimating any true association. Moreover, given that the median age at cancer diagnosis is 66 years according to the NCI’s Surveillance, Epidemiology, and End Results (SEER) Program, the relatively younger mean age of our cohort suggests that the follow-up period may not have been sufficient to capture the onset of malignancies—further contributing to potential underestimation of the association between microvascular complications and cancer risk [30].

One of the key strengths of our study is the use of a large, nationally representative cohort with longitudinal follow-up, providing a robust dataset for evaluating long-term cancer risk in T2D patients. The propensity score matching approach further strengthens our findings by balancing baseline characteristics and reducing potential confounders that could bias the analysis.

Our study has several limitations that should be considered. First, while our follow-up period was substantial, cancer development is a long-term process, and some early-stage cancers may not have been fully captured within the observation window. Second, despite adjusting for major risk factors, residual confounding from lifestyle factors such as diet, physical activity, and genetic predisposition may still influence cancer risk. Third, our reliance on ICD-coded diagnoses introduces the possibility of misclassification bias, as subclinical or undiagnosed cancers may not be recorded in the database. Fourth, this dataset does not provide serial blood glucose, haemoglobin A1C, or insulin levels, which may affect cancer development and progression [12,31]. Additionally, although NHIRD provides comprehensive health data, it lacks granular details on tumour subtypes, cancer staging, and treatment response, limiting our ability to analyse cancer progression in greater detail.

Future research should consider extending the follow-up duration and conducting similar analyses in large international cohorts to determine whether a latent effect of microvascular disease on cancer incidence emerges over time. Another important area of investigation is the role of macrovascular complications in cancer risk. While our study focused on microvascular disease, it remains unclear whether conditions such as coronary artery disease, cerebrovascular disease, or peripheral artery disease influence cancer outcomes differently. Finally, further studies should explore how microvascular disease affects cancer treatment outcomes, including chemotherapy response, drug toxicity, and surgical healing, as these insights could help optimize cancer care in diabetic patients.

## 5. Conclusions

In this population-based cohort study, we found that microvascular complications in T2D were not significantly associated with a higher risk of cancer or cancer-related mortality, suggesting that these vascular impairments may not be key drivers of tumourigenesis. However, given the observational nature of the study and potential diagnostic limitations—such as the lack of biochemical data and the possibility of cancer underdiagnosis—these findings should be interpreted with caution. In contrast, the strong dose-dependent association between microvascular disease and all-cause mortality highlights the substantial impact of these complications on long-term survival. Further research is warranted to refine risk stratification and guide the development of targeted interventions aimed at optimising vascular health and overall outcomes in patients with diabetes.

## Figures and Tables

**Figure 1 cancers-17-01760-f001:**
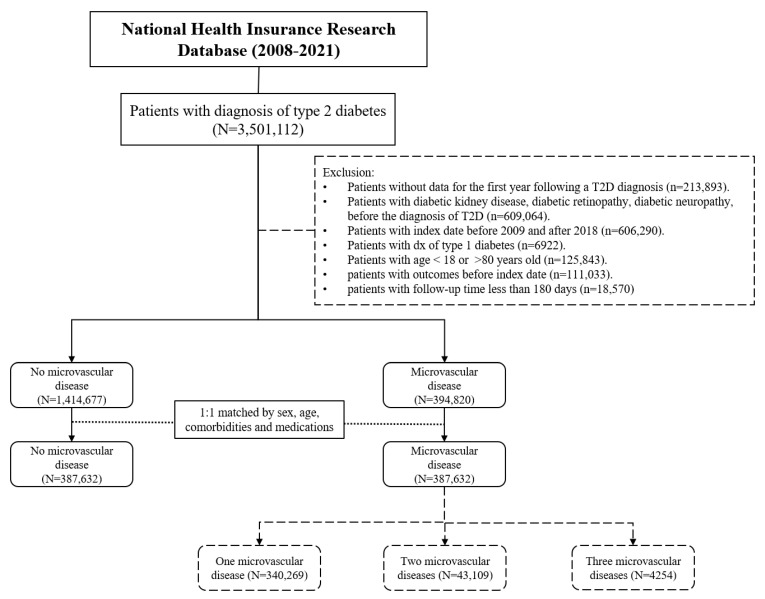
Patient selection flow chart.

**Figure 2 cancers-17-01760-f002:**
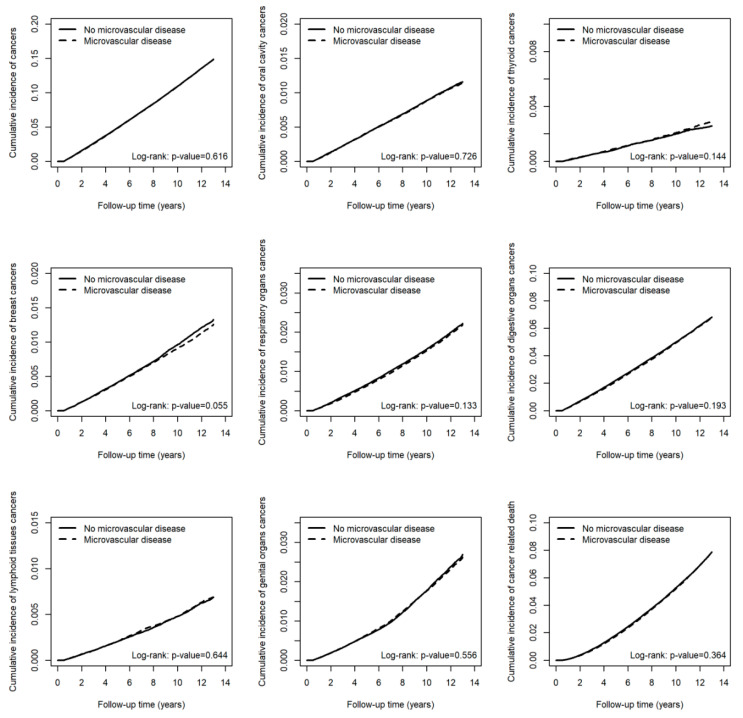
Cumulative incidence of cancers and cancer-related death in patients with and without microvascular complications.

**Figure 3 cancers-17-01760-f003:**
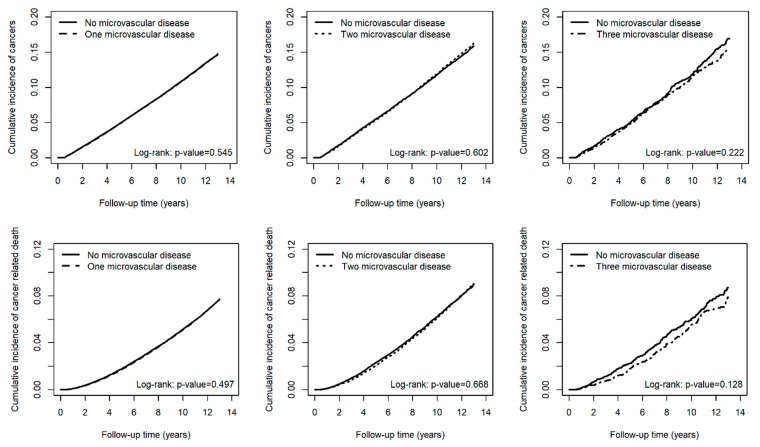
Cumulative incidence of all cancers and cancer-related death in matched patients without microvascular disease and with one, two, or three microvascular diseases.

**Table 1 cancers-17-01760-t001:** Hazard ratios and 95% confidence intervals for outcomes among matched patients without and with microvascular disease.

Exposure	All Cancers			cHR	(95% CI)	*p*-Value	aHR ^†^	(95% CI)	*p*-Value
	*n*	PY	IR						
No microvascular disease	38,335	3,316,979.77	11.56	1.00	(reference)	-	1.00	(reference)	-
Microvascular disease	37,380	3,255,697.38	11.48	1.00	(0.98, 1.01)	0.616	1.00	(0.98, 1.01)	0.695
Exposure	Oral cavity cancers			cHR	(95% CI)	*p*-value	aHR ^†^	(95% CI)	*p*-value
	n	PY	IR						
No microvascular disease	2999	3,418,211.78	0.88	1.00	(reference)	-	1.00	(reference)	-
Microvascular disease	2913	3,355,570.42	0.87	0.99	(0.94, 1.04)	0.726	0.99	(0.94, 1.05)	0.807
Exposure	Thyroid cancers			cHR	(95% CI)	*p*-value	aHR ^†^	(95% CI)	*p*-value
	n	PY	IR						
No microvascular disease	667	3,424,998.77	0.19	1.00	(reference)	-	1.00	(reference)	-
Microvascular disease	707	3,362,022.27	0.21	1.08	(0.97, 1.20)	0.144	1.07	(0.96, 1.19)	0.204
Exposure	Breast cancers			cHR	(95% CI)	*p*-value	aHR ^†^	(95% CI)	*p*-value
	n	PY	IR						
No microvascular disease	3279	3,413,642.23	0.96	1.00	(reference)	-	1.00	(reference)	-
Microvascular disease	3060	3,351,524.49	0.91	0.95	(0.91, 1.00)	0.055	0.95	(0.91, 1.00)	0.058
Exposure	Respiratory organ cancer			cHR	(95% CI)	*p*-value	aHR ^†^	(95% CI)	*p*-value
	n	PY	IR						
No microvascular disease	5462	3,418,239.84	1.60	1.00	(reference)	-	1.00	(reference)	-
Microvascular disease	5190	3,355,484.97	1.55	0.97	(0.94, 1.01)	0.133	0.97	(0.93, 1.01)	0.112
Exposure	Digestive organ cancers			cHR	(95% CI)	*p*-value	aHR ^†^	(95% CI)	*p*-value
	n	PY	IR						
No microvascular disease	17,305	3,384,166.79	5.11	1.00	(reference)	-	1.00	(reference)	-
Microvascular disease	16,709	3,323,144.91	5.03	0.99	(0.97, 1.01)	0.194	0.99	(0.97, 1.01)	0.286
Exposure	Lymphoid tissue cancers			cHR	(95% CI)	*p*-value	aHR ^†^	(95% CI)	*p*-value
	n	PY	IR						
No microvascular disease	1672	3,423,921.91	0.49	1.00	(reference)	-	1.00	(reference)	-
Microvascular disease	1662	3,361,202.11	0.49	1.02	(0.95, 1.09)	0.644	1.02	(0.95, 1.09)	0.651
Exposure	Genital organ cancers			cHR	(95% CI)	*p*-value	aHR ^†^	(95% CI)	*p*-value
	n	PY	IR						
No microvascular disease	6169	3,407,260.84	1.81	1.00	(reference)	-	1.00	(reference)	-
Microvascular disease	5958	3,344,958.26	1.78	0.99	(0.95, 1.03)	0.556	0.99	(0.96, 1.03)	0.597
Exposure	Cancer-related death			cHR	(95% CI)	*p*-value	aHR ^†^	(95% CI)	*p*-value
	n	PY	IR						
No microvascular disease	18,694	3,427,998.00	5.45	1.00	(reference)	-	1.00	(reference)	-
Microvascular disease	18,066	3,365,206.49	5.37	0.99	(0.97, 1.01)	0.364	1.00	(0.98, 1.02)	0.695
Exposure	All-cause mortality			cHR	(95% CI)	*p*-value	aHR ^†^	(95% CI)	*p*-value
	n	PY	IR						
No microvascular disease	88,676	3,427,998.00	25.87	1.00	(reference)	-	1.00	(reference)	-
Microvascular disease	103,547	3,365,206.49	30.77	1.20	(1.18, 1.21)	<0.001	1.21	(1.20, 1.22)	<0.001

PY = person-years; IR = incidence rate per 1000 person-years; cHR = crude hazard ratio; aHR, adjusted hazard ratio. ^†^ aHR = adjusted hazard ratio, adjusted by sex, age, obesity, smoking, alcohol-related disorders, comorbidities, antidiabetic drugs, and cardiovascular-related drugs, as listed in Appendix A.

**Table 2 cancers-17-01760-t002:** Hazard ratios and 95% confidence intervals for outcomes among matched patients without and with one to three microvascular diseases.

Exposure	All Cancers			cHR	(95% CI)	*p*-Value	aHR ^†^	(95% CI)	*p*-Value
	n	PY	IR						
No microvascular disease (N = 340,269)	33,405	2,923,248.64	11.43	1.00	(reference)	-	1.00	(reference)	-
One microvascular disease (N = 340,269)	32,494	2,865,864.22	11.34	1.00	(0.98, 1.01)	0.545	0.99	(0.98, 1.01)	0.39
No microvascular disease (N = 43,109)	4492	359,819.98	12.48	1.00	(reference)	-	1.00	(reference)	-
Two microvascular diseases (N = 43,109)	4490	356,397.57	12.60	1.01	(0.97, 1.05)	0.602	1.02	(0.98, 1.07)	0.274
No microvascular disease (N = 4254)	438	33,911.15	12.92	1.00	(reference)	-	1.00	(reference)	-
Three microvascular diseases (N = 4254)	396	33,435.58	11.84	0.92	(0.80, 1.05)	0.223	0.99	(0.86, 1.14)	0.862
Exposure	Oral cavity cancers			cHR	(95% CI)	*p*-value	aHR ^†^	(95% CI)	*p*-value
	n	PY	IR						
No microvascular disease (N = 340,269)	2607	3,012,368.23	0.87	1.00	(reference)	-	1.00	(reference)	-
One microvascular disease (N = 340,269)	2547	2,952,872.32	0.86	1.00	(0.95, 1.05)	0.946	1.00	(0.95, 1.06)	0.984
No microvascular disease (N = 43,109)	359	370,937.52	0.97	1.00	(reference)	-	1.00	(reference)	-
Two microvascular diseases (N = 43,109)	329	368,167.73	0.89	0.92	(0.80, 1.07)	0.303	0.93	(0.80, 1.08)	0.359
No microvascular disease (N = 4254)	33	34,906.02	0.95	1.00	(reference)	-	1.00	(reference)	-
Three microvascular diseases (N = 4254)	37	34,530.38	1.07	1.13	(0.71, 1.81)	0.605	1.15	(0.71, 1.87)	0.574
Exposure	Thyroid cancers			cHR	(95% CI)	*p*-value	aHR ^†^	(95% CI)	*p*-value
	n	PY	IR						
No microvascular disease (N = 340,269)	611	3,018,303.70	0.20	1.00	(reference)	-	1.00	(reference)	-
One microvascular disease (N = 340,269)	629	2,958,586.78	0.21	1.05	(0.94, 1.18)	0.367	1.04	(0.93, 1.16)	0.477
No microvascular disease (N = 43,109)	51	371,709.52	0.14	1.00	(reference)	-	1.00	(reference)	-
Two microvascular diseases (N = 43,109)	69	368,836.10	0.19	1.37	(0.95, 1.96)	0.092	1.38	(0.96, 1.99)	0.080
No microvascular disease (N = 4254)	5	34,985.55	0.14	1.00	(reference)	-	1.00	(reference)	-
Three microvascular diseases (N = 4254)	9	34,599.39	0.26	1.83	(0.61, 5.46)	0.279	1.76	(0.55, 5.64)	0.34
Exposure	Breast cancers			cHR	(95% CI)	*p*-value	aHR ^†^	(95% CI)	*p*-value
	n	PY	IR						
No microvascular disease (N = 340,269)	2870	3,008,467.58	0.95	1.00	(reference)	-	1.00	(reference)	-
One microvascular disease (N = 340,269)	2688	2,949,312.20	0.91	0.96	(0.91, 1.01)	0.11	0.96	(0.91, 1.01)	0.107
No microvascular disease (N = 43,109)	374	370,330.20	1.01	1.00	(reference)	-	1.00	(reference)	-
Two microvascular diseases (N = 43,109)	340	367,735.82	0.92	0.92	(0.79, 1.06)	0.242	0.91	(0.78, 1.05)	0.205
No microvascular disease (N = 4254)	35	34,844.45	1.00	1.00	(reference)	-	1.00	(reference)	-
Three microvascular diseases (N = 4254)	32	34,476.47	0.93	0.93	(0.57, 1.50)	0.757	0.99	(0.61, 1.61)	0.964
Exposure	Respiratory organ cancer			cHR	(95% CI)	*p*-value	aHR ^†^	(95% CI)	*p*-value
	n	PY	IR						
No microvascular disease (N = 340,269)	4772	3,012,458.33	1.58	1.00	(reference)	-	1.00	(reference)	-
One microvascular disease (N = 340,269)	4530	2,952,831.52	1.53	0.97	(0.93, 1.01)	0.17	0.97	(0.93, 1.01)	0.087
No microvascular disease (N = 43,109)	623	370,872.46	1.68	1.00	(reference)	-	1.00	(reference)	-
Two microvascular diseases (N = 43,109)	613	368,098.22	1.67	0.99	(0.89, 1.11)	0.912	1.02	(0.91, 1.14)	0.712
No microvascular disease (N = 4254)	67	34,909.05	1.92	1.00	(reference)	-	1.00	(reference)	-
Three microvascular diseases (N = 4254)	47	34,555.22	1.36	0.71	(0.49, 1.04)	0.078	0.89	(0.61, 1.30)	0.552
Exposure	Digestive organ cancers			cHR	(95% CI)	*p*-value	aHR ^†^	(95% CI)	*p*-value
	n	PY	IR						
No microvascular disease (N = 340,269)	15,022	2,982,745.40	5.04	1.00	(reference)	-	1.00	(reference)	-
One microvascular disease (N = 340,269)	14,448	2,925,036.74	4.94	0.98	(0.96, 1.01)	0.159	0.98	(0.96, 1.01)	0.142
No microvascular disease (N = 43,109)	2081	366,869.63	5.67	1.00	(reference)	-	1.00	(reference)	-
Two microvascular diseases (N = 43,109)	2080	363,960.89	5.71	1.01	(0.95, 1.07)	0.775	1.02	(0.96, 1.09)	0.452
No microvascular disease (N = 4254)	202	34,551.76	5.85	1.00	(reference)	-	1.00	(reference)	-
Three microvascular diseases (N = 4254)	181	34,147.28	5.30	0.91	(0.74, 1.11)	0.337	0.94	(0.76, 1.16)	0.552
Exposure	Lymphoid tissue cancers			cHR	(95% CI)	*p*-value	aHR ^†^	(95% CI)	*p*-value
	n	PY	IR						
No microvascular disease (N = 340,269)	1474	3,017,412.92	0.49	1.00	(reference)	-	1.00	(reference)	-
One microvascular disease (N = 340,269)	1431	2,957,979.29	0.48	0.99	(0.92, 1.07)	0.868	0.99	(0.92, 1.07)	0.811
No microvascular disease (N = 43,109)	178	371,546.45	0.48	1.00	(reference)	-	1.00	(reference)	-
Two microvascular diseases (N = 43,109)	214	368,630.99	0.58	1.21	(0.99, 1.48)	0.056	1.24	(1.02, 1.52)	0.033
No microvascular disease (N = 4254)	20	34,962.54	0.57	1.00	(reference)	-	1.00	(reference)	-
Three microvascular diseases (N = 4254)	17	34,591.82	0.49	0.88	(0.46, 1.69)	0.707	1.01	(0.52, 1.96)	0.968
Exposure	Genital organ cancers			cHR	(95% CI)	*p*-value	aHR ^†^	(95% CI)	*p*-value
	n	PY	IR						
No microvascular disease (N = 340,269)	5401	3,002,916.16	1.80	1.00	(reference)	-	1.00	(reference)	-
One microvascular disease (N = 340,269)	5290	2,943,288.52	1.80	1.01	(0.97, 1.04)	0.782	1.00	(0.97, 1.04)	0.856
No microvascular disease (N = 43,109)	704	369,563.22	1.90	1.00	(reference)	-	1.00	(reference)	-
Two microvascular diseases (N = 43,109)	625	367,172.57	1.70	0.90	(0.81, 1.00) *	0.048	0.91	(0.82, 1.02)	0.101
No microvascular disease (N = 4254)	64	34,781.46	1.84	1.00	(reference)	-	1.00	(reference)	-
Three microvascular diseases (N = 4254)	43	34,497.18	1.25	0.68	(0.46, 1.00) *	0.048	0.74	(0.49, 1.10)	0.131
Exposure	Cancer-related death			cHR	(95% CI)	*p*-value	aHR ^†^	(95% CI)	*p*-value
	n	PY	IR						
No microvascular disease (N = 340,269)	16,105	3,021,083.08	5.33	1.00	(reference)	-	1.00	(reference)	-
One microvascular disease (N = 340,269)	15,561	2,961,395.35	5.25	0.99	(0.97, 1.01)	0.497	0.99	(0.97, 1.02)	0.558
No microvascular disease (N = 43,109)	2372	371,912.77	6.38	1.00	(reference)	-	1.00	(reference)	-
Two microvascular diseases (N = 43,109)	2320	369,175.82	6.28	0.99	(0.93, 1.05)	0.668	1.01	(0.95, 1.07)	0.814
No microvascular disease (N = 4254)	217	35,002.15	6.20	1.00	(reference)	-	1.00	(reference)	-
Three microvascular diseases (N = 4254)	185	34,635.31	5.34	0.86	(0.71, 1.05)	0.129	0.99	(0.81, 1.22)	0.943
Exposure	All-cause mortality			cHR	(95% CI)	*p*-value	aHR ^†^	(95% CI)	*p*-value
	n	PY	IR						
No microvascular disease (N = 340,269)	74,983	3,021,083.08	24.82	1.00	(reference)	-	1.00	(reference)	-
One microvascular disease (N = 340,269)	84,043	2,961,395.35	28.38	1.15	(1.14, 1.16)	<0.001	1.16	(1.15, 1.17)	<0.001
No microvascular disease (N = 43,109)	12,300	371,912.77	33.07	1.00	(reference)	-	1.00	(reference)	-
Two microvascular diseases (N = 43,109)	17,214	369,175.82	46.63	1.41	(1.38, 1.45)	<0.001	1.42	(1.38, 1.45)	<0.001
No microvascular disease (N = 4254)	1393	35,002.15	39.80	1.00	(reference)	-	1.00	(reference)	-
Three microvascular diseases (N = 4254)	2290	34,635.31	66.12	1.66	(1.56, 1.78)	<0.001	1.71	(1.60, 1.83)	<0.001

PY = person-years; IR = incidence rate per 1000 person-years; cHR = crude hazard ratio; aHR, adjusted hazard ratio. ^†^ aHR = adjusted hazard ratio, adjusted by sex, age, obesity, smoking, alcohol-related disorders, comorbidities, antidiabetic drugs, and cardiovascular-related drugs, as listed in Appendix A. * *p* < 0.05.

## Data Availability

Study data are available from the National Health Insurance Research Database (NHIRD), published by the Taiwan National Health Insurance (NHI) Administration. The data used in this study cannot be made available in the paper, Appendix A, or a public repository due to the “Personal Information Protection Act” executed by the Taiwan government since 2012. Requests for data can be sent as a formal proposal to the NHIRD Office (https://dep.mohw.gov.tw/DOS/cp-5119-59201-113.html (accessed on 1 January 2025)) or by email to stsung@mohw.gov.tw.

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
