# Peer review of "Microvascular Complications and Cancer Risk in Type 2 Diabetes: A Population-Based Study"

_cancers, 2025, doi:10.3390/cancers17111760_

Round 1
Reviewer 1 Report
Comments and Suggestions for Authors
Interesting study, appropriate methods, well written. The only minor issue is following:
Authors show that microvascular T2D complications are not assoictaed with cancer and cancer mortality- this is importnat and novel, and would be enough to be published.
Another consulsion that these complications increase the all cause mortality is very simple, good known and logical, and so not as novel to be published alone.
I recommend to accept the article in the present form, but I still think that to show cancer and cancer mortality alone and not all cause mortality would get a better impression.
Author Response
Responses to the comments of Reviewer #1
Comments and Suggestions for Authors
Interesting study, appropriate methods, well written. The only minor issue is following:
Authors show that microvascular T2D complications are not associated with cancer and cancer mortality- this is important and novel, and would be enough to be published.
Response: We sincerely thank you for reviewing our manuscript and for your kind and encouraging feedback.
Another conclusion that these complications increase the all-cause mortality is very simple, good known and logical, and so not as novel to be published alone.
Response: We agree with your comment that the association between microvascular complications and increased all-cause mortality is well-established and clinically logical. We chose to retain this result as a form of positive outcome control, as it may help support the credibility of the findings in our study.
I recommend to accept the article in the present form, but I still think that to show cancer and cancer mortality alone and not all cause mortality would get a better impression.
Response: Thank you very much for your encouraging feedback and valuable suggestion. We agree that focusing solely on cancer incidence and cancer-specific mortality could sharpen the narrative. However, given that prior studies have shown discordance between cardiovascular and all-cause mortality, we felt it was important to present both cancer-related and all-cause mortality outcomes in our analysis. In our cohort as well, the association between microvascular complications and cancer-related mortality did not fully mirror that of all-cause mortality. Including both outcomes provides a more holistic view of the potential long-term impact of microvascular disease in patients with type 2 diabetes. We sincerely appreciate your insight.
Reviewer 2 Report
Comments and Suggestions for Authors
Thank you for submitting your work for consideration in Cancers
The manuscript discusses an important idea, which is the relationship between T2DM complications and the development of cancers
Here are my comments
-The introduction section needs to be clearer to express your idea better
-Methods section, this is the best part, especially figure 1
-The result section is well written, especially the table part. However, in Figure 2, the quality of the axis titles should be improved (I believe the resolution of Figure 2 is not of publication quality)
-Reference section, since this is a population study, I expect to see more references in this manuscript than only 22
-One of the points of strength of this study is the large population dataset used, and this is a significant point
-Where is the study limitation section?
-I believe figures should be increased than only 2, while tables are fine, so I highly recommend adding more figures to the study
-English needs to be revised, as a lot of typos were detected
Overall, the study is interesting but needs more revision
Comments on the Quality of English Language
-English needs to be revised, as a lot of typos were detected
Author Response
Responses to the comments of Reviewer #2
Comments and Suggestions for Authors
Thank you for submitting your work for consideration in Cancers
The manuscript discusses an important idea, which is the relationship between T2DM complications and the development of cancers
Response: Thank you for reviewing our manuscript and for your encouraging comments.
Here are my comments
-The introduction section needs to be clearer to express your idea better
Response: Thank you for your feedback. We have revised the Introduction section to improve clarity and better articulate the rationale and main objectives of our study. We hope the revised version more effectively communicates the context and significance of our research (pages 2-3).
-Methods section, this is the best part, especially figure 1
Response: We sincerely thank you.
-The result section is well written, especially the table part. However, in Figure 2, the quality of the axis titles should be improved (I believe the resolution of Figure 2 is not of publication quality)
Response: We have revised and improved the axis titles in Figure 2 for greater clarity.
-Reference section, since this is a population study, I expect to see more references in this manuscript than only 22
Response: We have increased the number of references to over 30.
-One of the points of strength of this study is the large population dataset used, and this is a significant point
Response: Thank you very much.
-Where is the study limitation section?
Response: The discussion of study limitations can be found on page 13.
-I believe figures should be increased than only 2, while tables are fine, so I highly recommend adding more figures to the study
Response: Thank you. We have included a total of three figures in the manuscript.
-English needs to be revised, as a lot of typos were detected
Response: The English has been revised and all typographical errors have been corrected.
Overall, the study is interesting but needs more revision
Response: Thank you for your feedback. We appreciate your interest in our study and have carefully revised the manuscript to address all concerns and improve its overall clarity and quality.
Comments on the Quality of English Language
-English needs to be revised, as a lot of typos were detected
Response: Thank you for your comment. We have thoroughly revised the manuscript for language and clarity, and it has been professionally edited by a native English-speaking editor through Wordvice. We hope the revised version addresses the concerns regarding language quality.

Reviewer 3 Report
Comments and Suggestions for Authors
The presented study analyzes the Taiwanese NHIRD database to investigate the relationship between microvascular complications in type 2 diabetes (T2D) and cancer mortality. The sample size is large and the research topic is interesting. However, the work also has some deficiencies.
1. The analysis performed by the authors does not include data on glycosylated hemoglobin or duration of hyperglycemia, which are key in the development of microvascular complications. I recommend including these data or taking it as a limiting factor.
2. Given the large number of study subjects included, it would be convenient to perform subanalyses, for example, stratifying by sex, age ranges or types of drugs used.
3. The introduction seems to me to be too brief. It would be advisable to go more deeply into the mechanisms that link cancer with cardiovascular complications. In addition, I believe that information is needed to show the lack of studies on this subject not only in Taiwan but also in other parts of the world.
4. In Table 2, a significant p was found regarding cancer of lymphatic tissue, but the authors do not discuss it or take it importance. Explain.
5. The conclusion given by the authors seems to me to be too blunt; the population studied, possible diagnostic biases, etc. should be considered.
6. Spelling errors.
7. Unify the references, in addition, there are some duplicated references such as number 4 and 21.
8. From my point of view there are few references, they could add some, for example, some meta-analyses.
Author Response
Responses to the comments of Reviewer #3
Comments and Suggestions for Authors
The presented study analyzes the Taiwanese NHIRD database to investigate the relationship between microvascular complications in type 2 diabetes (T2D) and cancer mortality. The sample size is large and the research topic is interesting. However, the work also has some deficiencies.
- The analysis performed by the authors does not include data on glycosylated hemoglobin or duration of hyperglycemia, which are key in the development of microvascular complications. I recommend including these data or taking it as a limiting factor.
Response: Thank you for your thoughtful comment. We agree that glycosylated hemoglobin and the duration of hyperglycemia are important factors in the development of microvascular complications. However, as our database does not contain information on glycosylated hemoglobin or blood glucose levels, we were unable to assess the duration of hyperglycemia. In line with your suggestion, we have now acknowledged this as a limitation of our study in the revised manuscript (page 13).
- Given the large number of study subjects included, it would be convenient to perform subanalyses, for example, stratifying by sex, age ranges or types of drugs used.
Response: Thank you for your valuable suggestion. We agree that stratified analyses by sex, age groups, or medication types could provide additional insights. However, given that the overall associations between microvascular complications and cancer incidence or cancer-related mortality were largely neutral, with no clear or significant trends, we did not pursue further subgroup analyses in this study. We appreciate your understanding and will consider such stratified analyses in future investigations where subgroup effects may be more apparent.
- The introduction seems to me to be too brief. It would be advisable to go more deeply into the mechanisms that link cancer with cardiovascular complications. In addition, I believe that information is needed to show the lack of studies on this subject not only in Taiwan but also in other parts of the world.
Response: Thank you for your insightful feedback. We have revised the Introduction to provide a more detailed discussion of the potential mechanisms linking cancer and cardiovascular complications. Additionally, we have included context on the limited research available on this topic, both in Taiwan and internationally, to better highlight the relevance and novelty of our study (pages 2-3).
- In Table 2, a significant p was found regarding cancer of lymphatic tissue, but the authors do not discuss it or take it importance. Explain.
Response: Thank you for your thoughtful reminder. We have addressed the significant p-value related to lymphatic tissue cancer observed in Table 2 and included a discussion of this finding in the manuscript (pages 6, 12).
- The conclusion given by the authors seems to me to be too blunt; the population studied, possible diagnostic biases, etc. should be considered.
Response: Thank you for your valuable comment. We have revised the Conclusion to present a more balanced interpretation of our findings, taking into account the characteristics of the study population and potential diagnostic biases. These considerations have also been acknowledged as limitations of this population-based study in the revised manuscript.
- Spelling errors.
Response: We have corrected all spelling errors in the manuscript.
- Unify the references, in addition, there are some duplicated references such as number 4 and 21.
Response: We sincerely appreciate your reminder. The duplicated references have been removed, and the reference list has been standardized for consistency.
- From my point of view there are few references, they could add some, for example, some meta-analyses.
Response: Thank you for your suggestion. We have incorporated additional references, including relevant meta-analyses, to strengthen the manuscript.
Round 2
Reviewer 3 Report
Comments and Suggestions for Authors
I agree with the revisions made by the authors.